# Pan-Cancer Analysis of the Roles and Driving Forces of RAB42

**DOI:** 10.3390/biom13010043

**Published:** 2022-12-26

**Authors:** Mingzhu Zheng, Wenhao Wang, Yulan Bu, Jing Liu, Jingang Ma, Rui Wang, Xiaomin Ren, Zhong Lu, Jiaqiu Li, Jinwei Cai

**Affiliations:** 1Department of Oncology, Affiliated Hospital of Weifang Medical University, School of Clinical Medicine, Weifang Medical University, Weifang 261031, China; 2Department of Radiotherapy, Affiliated Hospital of Weifang Medical University, Weifang 261031, China; 3Jinming Yu Academician Workstation of Oncology, Clinical Research Center, Affiliated Hospital of Weifang Medical University, Weifang 261031, China; 4Department of Oncology, People’s Hospital of Kecheng District, Quzhou 324000, China

**Keywords:** RAB42, pan-cancer, bioinformatics, biomarker, tumor immunity, super enhancer

## Abstract

RAB42 is a member of the RAS family. However, the roles and driving forces for RAB42 in tumors remain elusive. In this study, we performed a comprehensive pan-cancer analysis of the roles and regulatory mechanisms of RAB42 using bioinformatics and experiments. Online databases such as Sanger Box, ACLBI and TIDE were used to search for the expression levels, prognostic value and immune features of RAB42. We observed that RAB42 expression was upregulated in most tumors and was closely associated with poor prognosis. Enrichment analysis indicated that RAB42 was related to multiple biological functions, especially the immune process. RAB42 expression had a positive correlation with immune cell infiltration and immune checkpoint gene expression. RAB42 had a high predictive value for immunotherapy efficiency. Our study screened out susceptible drugs for the RAB42 protein by sensitivity analysis and virtual screening. Many key driver genes such as TP53 contributed to RAB42 expression. DNA methylation, super-enhancer and non-coding RNAs were the epigenetic factors responsible for RAB42 expression. In brief, RAB42 could serve as a diagnostic and prognostic biomarker in many tumor types. RAB42 might be a predictive biomarker and a new target for immunotherapy. Genetic and epigenetic factors were essential for RAB42 overexpression in tumors.

## 1. Introduction

The latest estimations by the World Health Organization showed that cancer is the second leading cause of death following cardiovascular disease in most countries [1]. The high incidence and mortality have placed a heavy burden on society. Although many treatments have been put into practice, the overall effectiveness of treatment is still limited due to the complexity of tumorigenesis. In recent years, immunotherapy, as an emerging treatment method, has brought light to tumor patients. The tumor microenvironment (TME) consists of immune cells, blood vessels and matrix composition [2]. TME has continuous interaction with cancer cells during tumor development. TME promotes tumor progression and immune evasion [3]. The inhibitors of immune checkpoint genes (ICIs), such as PD-1/PD-L1 and CTLA-4, are promising approaches for tumor patients [4]. They are progressively being used in clinical treatment as monotherapy or combined with other anticancer drugs [5]. However, these treatments have benefited only a small proportion of patients, so it is essential to discover new sensitive biomarkers and therapeutic targets for immunotherapy. 

In recent years, pan-cancer analysis has provided a new perspective for cancer research. Due to the advanced sequencing technique, a large amount of data can be obtained for comprehensive pan-cancer analysis [6]. The pan-cancer analysis is helpful to discover the similarities and differences among distinct tumor types. 

RAB42 is a member of the RAS oncogene family [7]. As a novel tumor biomarker, RAB42 is associated with poor prognosis in glioma [8]. It may promote glioma development through the VEGF signaling pathway [9]. Furthermore, RAB42 is correlated with immune infiltration in hepatocellular carcinoma [10]. However, the roles and regulatory mechanisms for RAB42 in other tumor types remain elusive. The aberrant expression of genes is mainly caused by genetic and epigenetic abnormalities, including DNA methylation, histone modification and non-coding RNA [11]. Recently, super enhancers (SEs), characterized by histone 3 lysine 27 acetylation (H3K27ac), have attracted extensive attention for their crucial roles in gene regulation. SEs are clusters of active enhancers, which enhance the expression of target genes [12]. 

In this research, we comprehensively analyzed the roles and regulatory mechanisms of RAB42 by pan-cancer datasets. We assessed RAB42 expression levels and prognostic value. Functional enrichment analysis was used to discover the underlying biological functions of RAB42. More attention was paid to analyze the link between RAB42 and immune features. Finally, we elucidated the factors responsible for the aberrant expression of RAB42 in tumors.

## 2. Materials and Methods

### 2.1. The SangerBox Database

The SangerBox database was used to evaluate RAB42 expression, prognostic value and immunological characteristics in pan-cancer. The abbreviations for different tumor types are shown in Appendix A. Through this database, we analyzed RAB42 expression in normal tissues and tumor tissues. Normal tissue data were from the Genotype-Tissue Expression (GTEx) database. Additionally, the tumor cell line data were from the Cancer Cell Line Encyclopedia (CCLE) database. By integrating data from The Cancer Genome Atlas (TCGA) and the GTEx database, we compared the differential RAB42 expression levels between tumor tissues and adjacent normal tissues. The influence of RAB42 expression on patient survival was assessed by univariate survival analysis and Kaplan–Meier analysis. By the SangerBox database, we obtained updated gene annotations for the Gene Ontology (GO) and the Kyoto Encyclopedia of Genes and Genomes (KEGG) pathway, and we performed functional enrichment analysis of RAB42-related genes. The gene set enrichment analysis (GSEA) tool was employed to conduct an enrichment analysis of the KEGG and HALLMARK pathways. We obtained Spearman’s correlation coefficient between RAB42 expression and immune infiltration score, tumor mutation burden (TMB) and microsatellite instability (MSI) in pan-cancer.

### 2.2. The Assistant for Clinical Bioinformatic Database

The RAB42 expression levels in 33 cancer types were analyzed by the “pan-cancer” module of the Assistant for Clinical Bioinformatic (ACLBI) database. The correlation between RAB42 expression and immune-infiltrating cells was calculated via XCELL, MCPCOUNTER and TIMER algorithms. The correlation between RAB42 and immune checkpoint gene expression was calculated by Spearman’s correlation coefficient.

### 2.3. The TIMER2.0 Database

The differential expression of RAB42 was analyzed by the “Gene_DE” module of the TIMER 2.0 database [13]. Wilcoxon’s test was performed to examine the statistical significance.

### 2.4. The UALCAN Database

The “TCGA” module of the UALCAN database was available for comparing RAB42 mRNA levels between tumor and normal tissues [14]. We assessed RAB42 expression based on TP53 mutation status and promoter methylation levels in pan-cancer.

### 2.5. The TISCH Database

The TISCH database integrates a large amount of single-cell RNA sequencing data aiming to characterize TME [15]. The distribution of RAB42 in different cells was evaluated using the “Dataset” module.

### 2.6. The TISIDB Database

As an online portal, TISIDB [16] collects clinical data from the TCGA database. The “Clinical” module was employed to explore the association between RAB42 expression and tumor stages. The “Immunotherapy” module provided the differential expression of RAB42 in responders and non-responders treated with immunotherapy.

### 2.7. The GEPIA2 Database

The “Stage Plot” pane of the GEPIA2 database [17] provided box plots of RAB42 expression at different tumor stages. The correlation between RAB42 and BRD4 expression in STAD was analyzed by the “Correlation Analysis” module.

### 2.8. The TIDE Database

The TIDE database collects extensive data from published immune checkpoint blockade (ICB) clinical studies [18]. We used the "Biomarker Evaluation" module to compare RAB42 with other biomarkers. AUC value is employed to test the predictive performance of biomarkers for ICB response in distinct treatment cohorts. The range of AUC is 0–1, and a larger value means better predictive performance. The “Query Gene” module was available for analyzing the association between RAB42 expression and the function of CTL (Cytotoxic T cell). Additionally, the prognosis of patients exposed to different immune checkpoint inhibitors was predicted.

### 2.9. STRING Online Tool

The STRING online tool assembles published available sources of protein-protein interaction (PPI) information [19]. The “minimum required interaction score” was medium confidence (0.400), and the “max number of interactors to show” was no more than 50 interactors. Other parameters were left as defaults.

### 2.10. The GeneMANIA Database

The GeneMANIA database is an online tool for biological network integration and gene function prediction [20]. The interaction networks of RAB42-related genes were mapped. We set “max resultant genes” to 80 and “max resultant attributes” to 10.

### 2.11. Screening Susceptible Drugs 

We downloaded the RNA-seq data and the NCI-60 drug data using the CellMiner database [21], and selected FDA-approved drugs or clinical trial drugs. Then, the R packages “impute”, “limma”, “ggplot2” and “ggpubr” were used for data processing and visualization. *p* < 0.05 was the filter condition for RAB42-related drugs. The Gene Set Cancer Analysis (GSCA) data analysis platform incorporates relevant data about pharmacogenomics from the GDSC and CTRP databases [22]. The "Drug" module was used to predict the sensitive drugs associated with RAB42.

### 2.12. Molecular Docking

The RAB42 protein structure was obtained from the AlphaFold database [23]. The chemical structure of drugs was downloaded from the PubChem database [24]. By using the GHECOM algorithm, we predicted possible drug-binding sites on the RAB42 protein [25]. Finally, DOCK 6.9 and PyMol software were employed for docking conformation. Interaction forces between amino acid residues of the RAB42 protein and drugs were visualized by Ligplus software.

### 2.13. The cBioPortal Database

The cBioPortal database can be used to analyze gene mutations. We used the “Cancer Type Summary” module to analyze RAB42 alteration frequency [26]. The “Mutations” module provided a mutational landscape of the cross-protein structural domains of RAB42. With the “Plots” module, we calculated the mutation count in different cancer types.

### 2.14. Cells

Human gastric cancer cell lines SGC7901 and BGC823 were used in our study. Additionally, these cell lines were purchased from the cell bank of the Chinese Academy of Sciences. Cells were cultured in RPMI1640 medium supplemented with 10% fetal bovine serum at 37 °C in an incubator containing 5% CO_2_. Cell culture dishes/plates and centrifuge tubes were purchased from NEST Biotechnology Co. Ltd., Jiangsu, China

### 2.15. siRNA Transfection

Cells were plated in 6-well plates and transfected with siRNA which was synthesized by Genepharma. After transfection for 48 h, RNA was extracted to identify the effectiveness of siRNA transfection. The siRNA sequences for BRD4 are shown below: 

BRD4-siRNA1#: S: CCGUGAUGCUCAGGAGUUUTT, AS: AAACUCCUGAGCAUCACGGTT; BRD4-siRNA2#: S: AGCUGAACCUCCCUGAUUATT, AS: UAAUCAGGGAGGUUCAGCUTT.

### 2.16. RNA Extraction and Quantitative Real-Time PCR (qRT-PCR) 

Cells in 6-well plates were treated with inhibitor JQ1-1uM and I-BET-762-2uM for 24 h. Then, total RNA was extracted from cells using the Trizol reagent. The qPCR assays were performed as reported previously [27]. a-Tubulin was used as the normalized endogenous control. The primer sequences used in this study are shown below: 

RAB42-F: CGAGCGCTTCAGGTGCA, R: GGTCACTCTTGTGGCCAACC; a-Tubulin-F: GAAGCAGCAACCATGCGTGA, R: AAGGAATCATCTCCTCCCCCA.

### 2.17. Chromatin Immunoprecipitation-qPCR

The chromatin immunoprecipitation (ChIP) assays were performed as reported previously [28]. H3K27ac antibody (ab177178, Abcam, Cambridge, UK) and BRD4 antibody (A301-985A100, Bethyl Laboratories, Montgomery, TX, USA) were used. The primer sequences of the RAB42 super-enhancer region are shown below: 

RAB42-H3K27ac-F: CAGTCCGAAGCGAGAGAAGG, R: GTGAATGTGGCACGGAGGA.

### 2.18. H3K27ac ChIP-seq Tracks

The ENCODE database incorporates the data of human and mouse gene annotations [29]. We downloaded H3K27ac ChIP-seq data of cell lines using ENCODE. The H3K27ac tracks were visualized by WashU Epigenome Browser.

### 2.19. Construction of ceRNA Network

The prediction of miRNAs targeting RAB42 was performed using online databases, including miRDB, miRWalK, miRtarBase and TargetScan [30,31,32,33]. The StarBase database [34] was used for the prediction of the potential long non-coding RNA (lncRNA) and circular RNA (circRNA) binding to RAB42-targeting miRNAs. The filtering criteria were as follows. “CLIP-Data” of lncRNA and circRNA was strict stringency. “Degradome data” of circRNA was high stringency. Cytoscape software was applied to visualize the ceRNA networks. We also analyzed the expression levels of these miRNAs/lncRNAs and their correlation with RAB42 in tumors. FDR is less than 0.05. The fold change is greater than 2.

### 2.20. Statistical Analysis

A two-sided Student’s *t*-test was used to assess statistical significance by using GraphPad Prism 7 software. If the results did not have the same SD, then we used a *t*-test with Welch’s correction. A *p* value of less than 0.05 was considered statistically significant. * *p* < 0.05; ** *p* < 0.01; *** *p* < 0.001; **** *p* < 0.0001.

## 3. Results

### 3.1. Expression Analysis of RAB42 in Pan-Cancer

RAB42 expression levels were analyzed in 31 normal tissues (Appendix A), showing the highest expression in spleen and the lowest in blood. We evaluated RAB42 expression in tumor cell lines from 21 tumor tissues (Appendix A). RAB42 was differentially expressed among tumor cell lines and the highest expression levels of RAB42 were found in kidney cancer cell lines. By comparing RAB42 mRNA levels in tumor and normal tissues using the SangerBox, ACLBI, TIMER2.0 and UALCAN databases, we observed the upregulated expression of RAB42 in most tumor tissues (Figure 1A,B and Appendix A). In particular, nine tumor tissues, including bladder urothelial carcinoma (BLCA), cholangiocarcinoma (CHOL), esophageal carcinoma (ESCA), glioblastoma multiforme (GBM), head and neck squamous cell carcinoma (HNSC), kidney renal clear cell carcinoma (KIRC), kidney renal papillary cell carcinoma (KIRP), lung adenocarcinoma (LUAD) and stomach adenocarcinoma (STAD), exhibited a remarkable upregulation of RAB42 expression in all of the above databases. In addition, RAB42 expression was observed to be closely correlated with tumor stages (Appendix A). However, this correlation is not always towards overexpression. In KIRP and liver hepatocellular carcinoma (LIHC), RAB42 was downregulated in more advanced stages. Next, the heterogeneity of RAB42 expression in diverse tumors was assessed using single-cell sequencing data (Appendix A). The expression of RAB42 in CHOL was most enriched in monocyte/macrophage. However, RAB42 was most enriched in fibroblast of KIRC and dendritic cell of STAD. These results showed variations of RAB42 expression in different tumor types.

### 3.2. Correlation between RAB42 Expression and Prognosis in Pan-Cancer

We wondered whether there is a potential correlation between RAB42 expression and prognosis. The hazard ratio (HR) and the *p* value of the four survival indicators of 33 tumors were calculated by univariate survival analysis using the SangerBox database (Appendix A). Figure 2 shows that RAB42 expression was tightly correlated to overall survival (OS) in adrenocortical carcinoma (ACC), colon adenocarcinoma (COAD), brain lower-grade glioma (LGG), LIHC, mesothelioma (MESO), ovarian serous cystadenocarcinoma (OV), uterine corpus endometrial carcinoma (UCEC) and uveal melanoma (UVM). Additionally, it was also related to disease-specific survival (DSS) in ACC, kidney chromophobe (KICH), KIRC, LGG, MESO, OV, pancreatic adenocarcinoma (PAAD), UCEC and UVM. Figure 3 indicates that RAB42 expression was closely associated with disease-free interval (DFI) in cervical squamous cell carcinoma and endocervical adenocarcinoma (CESC), LGG, PAAD and STAD. Additionally, it was also associated with progression-free interval (PFI) in CESC, KICH, KIRC, LGG, lung squamous cell carcinoma (LUSC), MSEO, PAAD, pheochromocytoma and paraganglioma (PCPG), prostate adenocarcinoma (PRAD), UCEC and UVM. The above outcomes indicated that RAB42 expression was apparently correlated to prognosis.

### 3.3. Functional Enrichment Analysis for RAB42

The functional enrichment analysis was performed to explore the oncogenic roles of RAB42. Appendix A illustrates the PPI networks of RAB42 by STRING online tool. The interaction networks of RAB42 were visualized via the GeneMANIA database (Appendix A). By the SangerBox database, we performed GO and KEGG pathway analysis for these RAB42-related genes (Figure 4). Notably, biological process (BP) revealed that RAB42 was involved in some immune pathways, including leukocyte activation, immune system process and immune response. In addition, the SangerBox database used lung adenocarcinoma (LUAD) data from the TCGA database to classify samples into high and low groups based on RAB42 expression. Enrichment of the KEGG and HALLMARK pathways in the high and low expression groups was analyzed using GSEA. It was found that the high-expressed RAB42 group enriched natural killer cell-mediated cytotoxicity and the Toll-like receptor signaling pathway in KEGG terms (Figure 5A). In the HALLMARK terms, the enriched pathways were IL2 STAT5 signaling, TNFA signaling via NFKB, interferon-gamma response and PI3K AKT mTOR signaling (Figure 5B). The results indicated that RAB42 was related to multiple biological functions, especially the immune process.

### 3.4. Correlation Analysis between RAB42 Expression and Immune Features

Considering that the functions of RAB42 may be related to immunity, we first calculated Spearman’s correlation coefficient between RAB42 expression and immune cell infiltration. All three algorithms revealed that RAB42 expression had a positive correlation with immune cell infiltration among the majority of cancer types (Figure 6A and Appendix A). Furthermore, RAB42 expression was positively related to Immune Score, Stromal Score and ESTIMATE Score in most of the tumor types (Figure 6B and Appendix A). Whether RAB42 has a predictive value for immunotherapy efficiency? As expected, we confirmed that RAB42 expression was positively correlated with immune checkpoint gene expression in most tumor types (Figure 7A). Additionally, RAB42 expression was observed to be positively related to MSI in BLCA, COAD and Sarcoma (SARC), while negatively related to MSI in HNSC, LUSC, skin cutaneous melanoma (SKCM), STAD and testicular germ cell tumors (TGCT) (Appendix A). Additionally, a positive correlation between RAB42 expression and TMB was exhibited in breast invasive carcinoma (BRCA), COAD, LGG, LUAD, OV and SARC, while a negative correlation was found in HNSC, SKCM, thyroid carcinoma (THCA) and thymoma (THYM) (Appendix A). More importantly, the TISIDB database indicated that RAB42 expression was higher in responders than non-responders of urothelial cancer treated with atezolizumab, a PD-L1 inhibitor (Figure 7B). The TIDE database showed that the patients with high-expressed RAB42 had a better prognosis treated with PD1 or CTLA4 inhibitors in melanoma (dbGaP Study Accession: phs000452.v3.p1) (Figure 7C). Notably, RAB42 had a higher predictive value for immunotherapy than other biomarkers (Figure 7D and Appendix A). Therefore, RAB42 might be a new predictive biomarker of immunotherapy efficacy. Interestingly, the TIDE database data showed that RAB42 expression was negatively correlated to the function of CTL (Cytotoxic T cell), implying its potential as a new target for immunotherapy (Appendix A).

### 3.5. Screening Susceptible Drugs for the RAB42 Protein

We used the CellMiner database to screen out 49 drugs associated with RAB42 expression (Appendix A). The scatter plots of the top 16 drugs in terms of correlation coefficient are shown in Figure 8. Additionally, we predicted many drugs sensitive to RAB42 by the GDSC and CTRP datasets in the GSCA database (Figure 9A,B). The top-ranked drugs were MG-132 and methotrexate in the CTRP and CDSC datasets, respectively. We intersected the two groups of drugs and obtained three drugs including AT-7519, BI-2536 and Dinaciclib for further analysis. The 3D structure of the RAB42 protein and potential binding sites are displayed in Figure 9C–E. The docking pocket is an area with a binding site. The largest volume of the pocket is usually selected. Additionally, the blue spheres are used to fill the cavity on the surface of the protein. We found a docking pocket with a maxi-mum volume of 1442 Å3 and set a box to wrap sphere. Appendix A shows the docking scores of these three drugs. We chose pose 1 of AT-7519, pose 2 of BI-2536 and pose 1 of Dinaciclib to display the 3D docking conformation with the RAB42 protein (Figure 9F–H). Figure 9I–K illustrates the interaction forces between amino acid residues of the RAB42 protein and drugs in a two-dimensional view.

### 3.6. The Effect of Genetic Factors on RAB42 Expression

Subsequently, we dug into the reasons for RAB42 overexpression in tumors. Firstly, we analyzed the alteration frequency of RAB42 in pan-cancer. The alteration frequency of RAB42 was less than 3% in most tumor types (Appendix A). We compared mutation counts between tumors, and estimated the mutant types and copy-number alterations of RAB42 (Appendix A). Next, we explored the driver genes responsible for RAB42 expression through the TCGA Portal database. We observed that many key driver genes, such as APC, TP53, KRAS, PTEN and CTNNB1, contributed to RAB42 expression in COAD, KIRC, LIHC and LUAD (Figure 10A). Extraordinarily, RAB42 expression was obviously relevant with TP53 mutation status (Figure 10B).

### 3.7. The Effect of Epigenetic Factors on RAB42 Expression

Other than genetic changes, aberrant gene expression could be the consequence of epigenetic abnormalities, including DNA methylation, histone modification and non-coding RNA. CpG island regions were predicted using the Sequence Manipulation Suite database (Appendix A). Positive correlations were found between RAB42 expression and methyltransferases in many tumors (Appendix A). Additionally, the promoter methylation levels of RAB42 decreased clearly in HNSC, KIRC, LIHC, TGCT and UCEC where RAB42 was overexpressed (Appendix A). On the contrary, the promoter methylation levels of RAB42 increased in COAD, LUSC, PRAD and THCA where RAB42 was not overexpressed. Recently, H3K27ac, as a marker for super enhancer, is the most studied histone modification. We identified abundant H3K27ac signal in the RAB42 gene loci in different tumor cells by ChIP-seq data from the ENCODE database (Figure 11A). QPCR assays suggested that RAB42 expression was upregulated in gastric cancer cells (Figure 11B). In accordance with the ChIP-seq data, a significant enrichment of H3K27ac to RAB42 gene loci was confirmed by ChIP-qPCR in SGC7901 and BGC823 (Figure 11C). Furthermore, RAB42 expression had a positive correlation with the expression of H3K27ac signal reader-BRD4 in STAD (Figure 11D). ChIP-qPCR assays indicated that BRD4 was also bound to RAB42 gene loci (Figure 11E). Inhibition of the BET domain or knockdown of BRD4 resulted in a decrease in the expression of RAB42 (Figure 11F,G). These data revealed that this super enhancer may be involved in the regulation of RAB42 expression. Finally, we predicted the potential ceRNA regulatory network for RAB42. We selected nine miRNAs as the potential target miRNAs of RAB42 by comparing multiple online databases. By the StarBase database, the miRNA–lncRNA and miRNA–circRNA interaction networks are illustrated in Appendix A, contributing to further elucidating the regulatory mechanism of RAB42. By the CancerMIRNome database, the expression levels of nine miRNAs in pan-cancer are shown in Appendix A. We conducted the co-expression analysis of miRNA-RAB42 interactions. Interestingly, their expression levels were negatively correlated in many tumors (Figure 12). The expression levels and the correlation with the RAB42 expression of the predicted lncRNA were analyzed by the StarBase database. The top five tumor types are shown in Appendix A. 

## 4. Discussion

As a protein-coding gene of the RAS oncogene family, RAB42 is involved in degenerated human discs and choroideremia disease [7,35]. However, there are few reports on the roles of RAB42 in tumors. In this paper, we tried to elaborate on its roles in tumors by pan-cancer analysis, which is beneficial for comparing the similarity and difference among distinct tumor types. This paper illustrated that RAB42 expression was higher in various tumors, especially in nine tumor tissues, than in adjacent normal tissues. Our results also demonstrated that RAB42 expression was correlated with tumor pathologic stages. Therefore, the upregulation of RAB42 may be involved in tumorigenesis and RAB42 could be considered as a specific diagnostic biomarker. Furthermore, RAB42 may have important clinical implications by indicating prognosis. A previous study confirmed that RAB42 overexpression served as an indicator of poor prognosis for glioma patients [9]. Our research identified that RAB42 was intimately tied to the prognostic indicators, including OS, DSS, DFI and PFI. RAB42 overexpression was an unfavorable prognostic indicator in a number of tumors. Therefore, RAB42 may also be recognized as a reliable prognostic biomarker. 

Unexpectedly, the functional enrichment analysis revealed that RAB42 was highly related to the immune process. GSEA pathway analysis verified that RAB42 expression was associated with immune-related pathways. Further findings presented a significant positive association between RAB42 expression and immune-infiltrating cells. This result was consistent with the latest study that RAB42 was a biomarker correlated with immune infiltration in hepatocellular carcinoma [10]. It has been confirmed that tumor progression and treatment resistance are affected by the immune-infiltrating cells. For example, macrophages respond to environmental signals with unique functions and are involved in tumor metastasis [36]. Tregs might promote tumor progression by inhibiting the antitumor immune response. The analysis of the correlation between RAB42 expression and immune cells implied a possible interaction between RAB42 and the tumor microenvironment. Inhibition of PD-1/PD-L1 has attracted significant attention as a new ideal treatment. Immune checkpoint inhibitors have been used for clinical therapy and have got a good therapeutic effect [37,38]. Interestingly, positive correlations between RAB42 and immune checkpoint gene expression existed in the majority of tumors. TMB is a surrogate for tumor neoantigen burden [39] and MSI is a biomarker for defective mismatch repair [40]. As new biomarkers, both of them contribute to predicting the immunotherapeutic effect [41]. Interestingly, we confirmed that RAB42 expression had positive correlations with TMB and MSI in COAD. Additionally, we discovered that RAB42 had a higher predictive value for immunotherapy than other biomarkers. Approximately 15% of colorectal cancers belong to MSI type [42]. However, not all MSI type colorectal cancer patients could benefit from immunotherapy. Therefore, RAB42 might be a powerful factor to assess whether colorectal cancer patients are absolutely suitable for immunotherapy. Furthermore, our study identified that the function of CTL (Cytotoxic T cell) was subject to RAB42 expression, providing a possibility that RAB42 may induce immune escape by impairing the function of T cells. Therefore, RAB42 has the potential to be a new target for immunotherapy. Anti-PD1/PD-L1 therapy failed in many theoretically suitable patients because of many factors such as exosomal PD-L1 [43]. Thus, whether targeting RAB42 could achieve a better immunotherapeutic effect? This hypothesis needs specific assays to validate. Here, our study predicted sensitivity drugs targeting the RAB42 protein by sensitivity analysis and virtual screening. Interestingly, among them, BI 2536 has demonstrated its antitumoral effect by accumulating CD8+ T cells in ovarian cancer [44]. This result was coincidentally consistent with our hypothesis. Of course, the effect and toxicity of these drugs in vivo assays need to be further verified. Maybe targeting RAB42 is a new promising strategy for cancer therapy. 

Further intensive studies of carcinogenic mechanisms are of great significance for targeting RAB42. The progressive accumulation of mutations might lead to loss of function and tumor formation [45]. However, RAB42 generally showed low mutational frequencies in pan-cancer. While many key driver genes contributed to RAB42 expression such as TP53. TP53 is an important tumor suppressor gene and its mutation leads to the rapid progression of many malignancies [46]. Analysis of TP53 mutation status in tumors is helpful in guiding tumor treatment [47]. Moreover, our research found that epigenetic alterations were also indispensable for RAB42 expression, especially the H3K27ac-mediated super enhancers (SEs). The definition of SEs is clusters of enhancers, where enriched with amounts of master regulators and mediators [48]. SEs are able to drive higher levels of transcriptional expression [49]. SEs are thought to be necessary to maintain cancer cell identity and play a key role in many tumors [50]. Studies have confirmed that SEs promote tumor progression such as colorectal cancer and squamous cancer [51,52]. However, less is known about SEs in gastric cancer. Our experiments confirmed the presence of super enhancers in the gene loci of RAB42 and verified the effect of super enhancers on RAB42 expression. Except for super enhancers, DNA methylation is intimately involved in regulating gene expression and tumorigenesis [53]. Increased promoter methylation levels tend to be negatively correlated with gene expression levels [54]. These results were in accordance with the methylation status in the RAB42 promoter region. In RAB42 overexpressed tumors, the methylation levels of RAB42 decreased clearly. Inversely, in some RAB42 non-overexpressed tumors, the methylation levels of RAB42 increased. Thus, low DNA methylation levels may contribute to RAB42 overexpression in tumors. Additionally, ceRNA regulatory network, crosstalk between RNAs, is extensively involved in the tumor process [55]. In the present study, we constructed a RAB42-related ceRNA network, providing the foundation for further exploring the regulatory mechanisms of RAB42. All of these findings demonstrated that genetic and epigenetic factors were essential for the upregulation of RAB42 expression in tumors. Our study also has certain limitations. Although RAB42 was analyzed comprehensively by bioinformatics, more in-depth studies are still needed to verify our opinions. Additionally, exact regulatory mechanisms need further validation by experiments.

## 5. Conclusions

RAB42 could serve as a diagnostic and prognostic biomarker in many tumor types. RAB42 might be a predictive biomarker and a new target for immunotherapy. Genetic and epigenetic factors were essential for RAB42 overexpression in tumors.

## Figures and Tables

**Figure 1 biomolecules-13-00043-f001:**
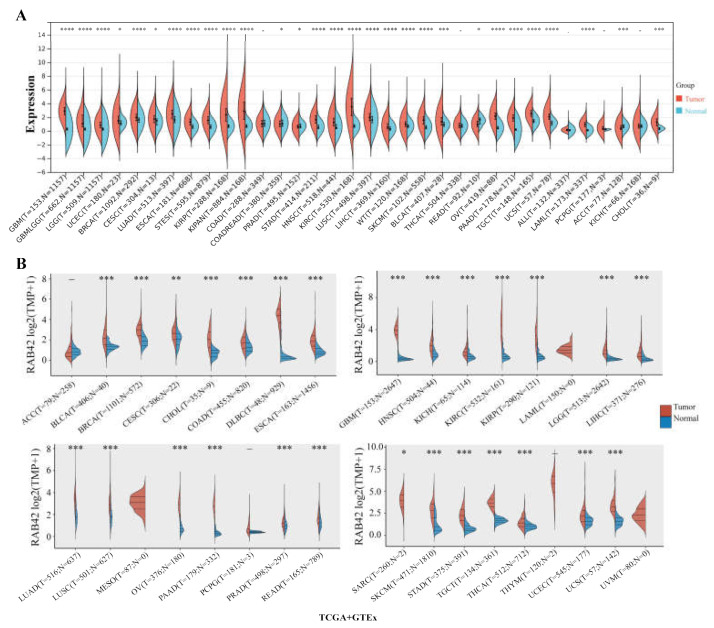
Expression analysis of RAB42 in pan-cancer. Comparison of RAB42 expression between normal and tumor tissues in the SangerBox (**A**) and ACLBI (**B**) databases. * *p* < 0.05, ** *p* < 0.01, *** *p* < 0.001, and **** *p* < 0.0001.

**Figure 2 biomolecules-13-00043-f002:**
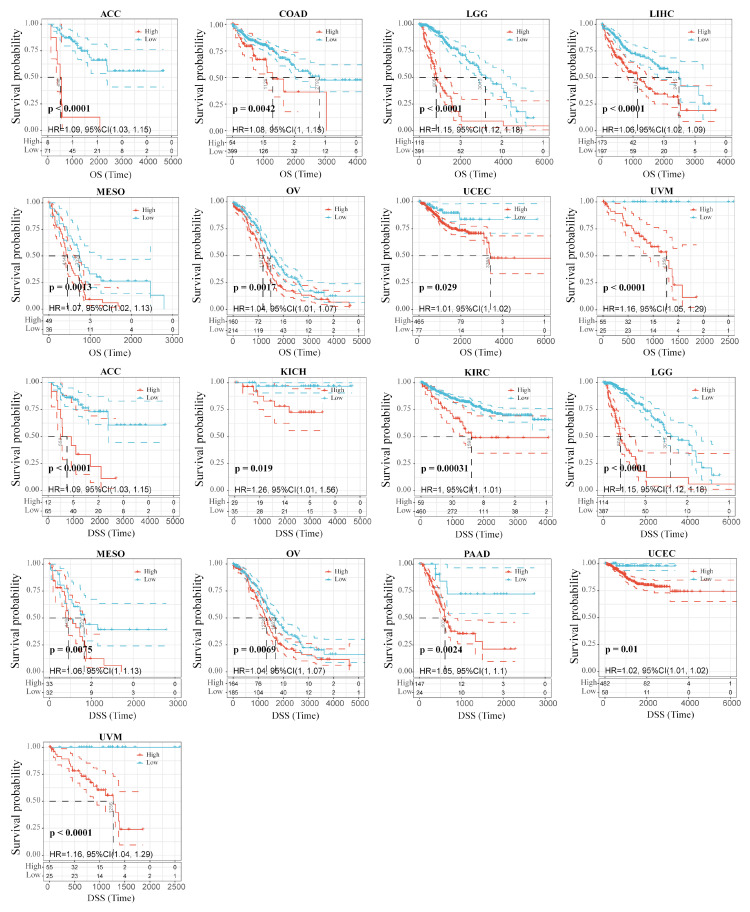
Correlation between RAB42 expression and OS and DSS.

**Figure 3 biomolecules-13-00043-f003:**
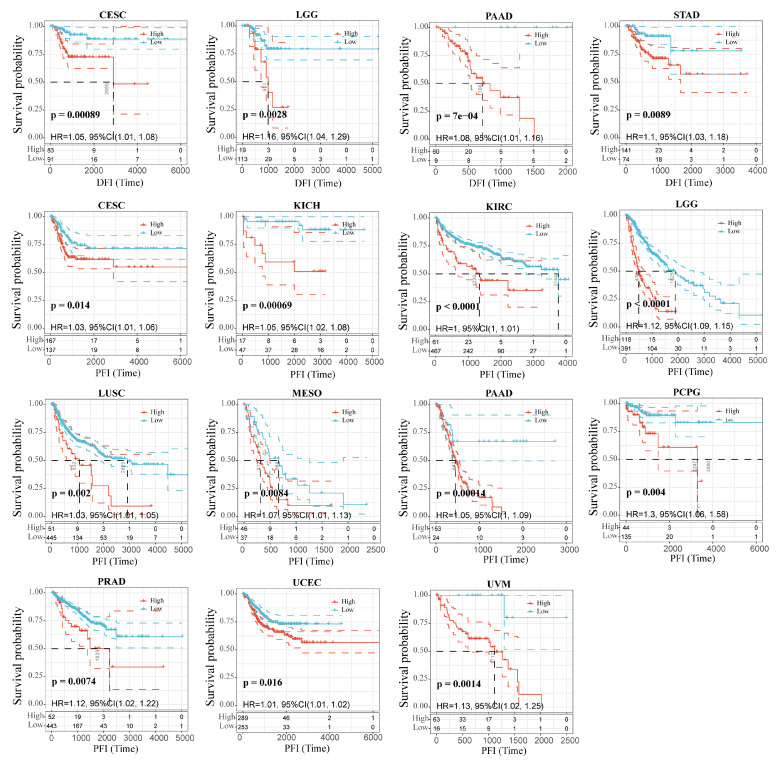
Correlation between RAB42 expression and DFI and PFI.

**Figure 4 biomolecules-13-00043-f004:**
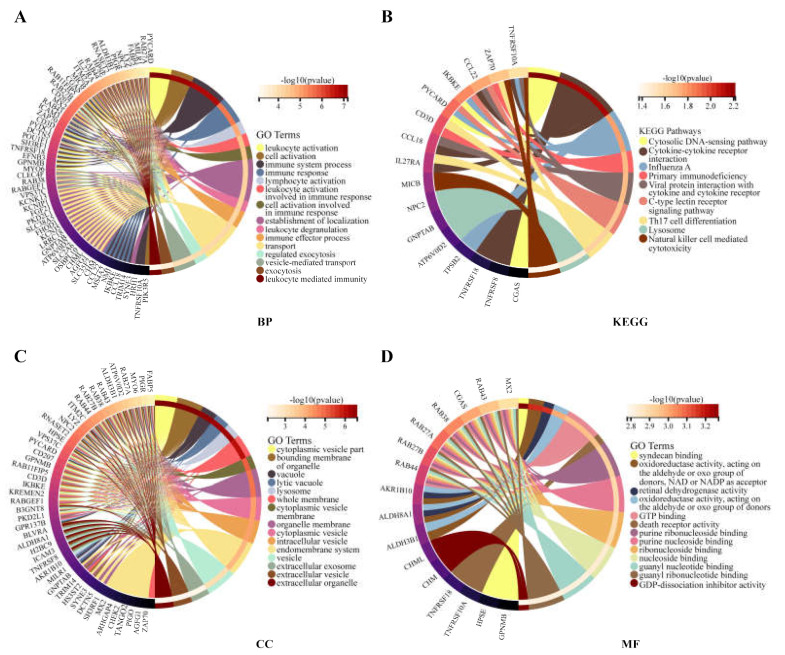
GO and KEGG analysis for RAB42-related genes. (**A**) BP, biological process. (**B**) KEGG, Kyoto encyclopedia of genes and genomes. (**C**) CC, cellular component. (**D**) MF, molecular function.

**Figure 5 biomolecules-13-00043-f005:**
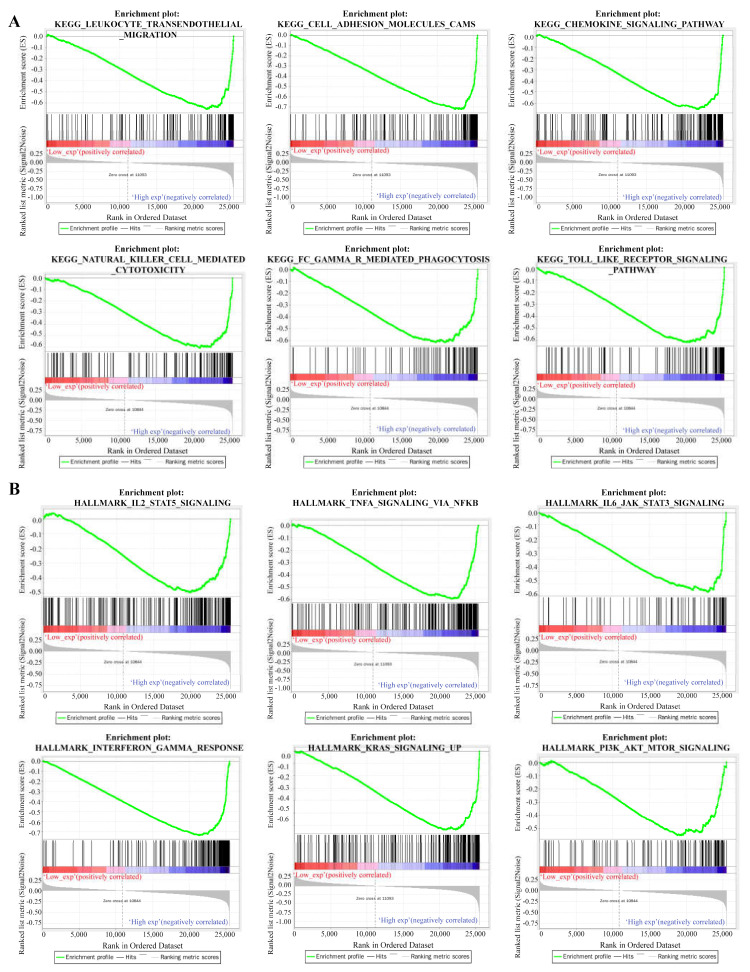
Gene set enrichment analysis (GSEA) for RAB42. KEGG pathways (**A**) and HALLMARK pathways (**B**).

**Figure 6 biomolecules-13-00043-f006:**
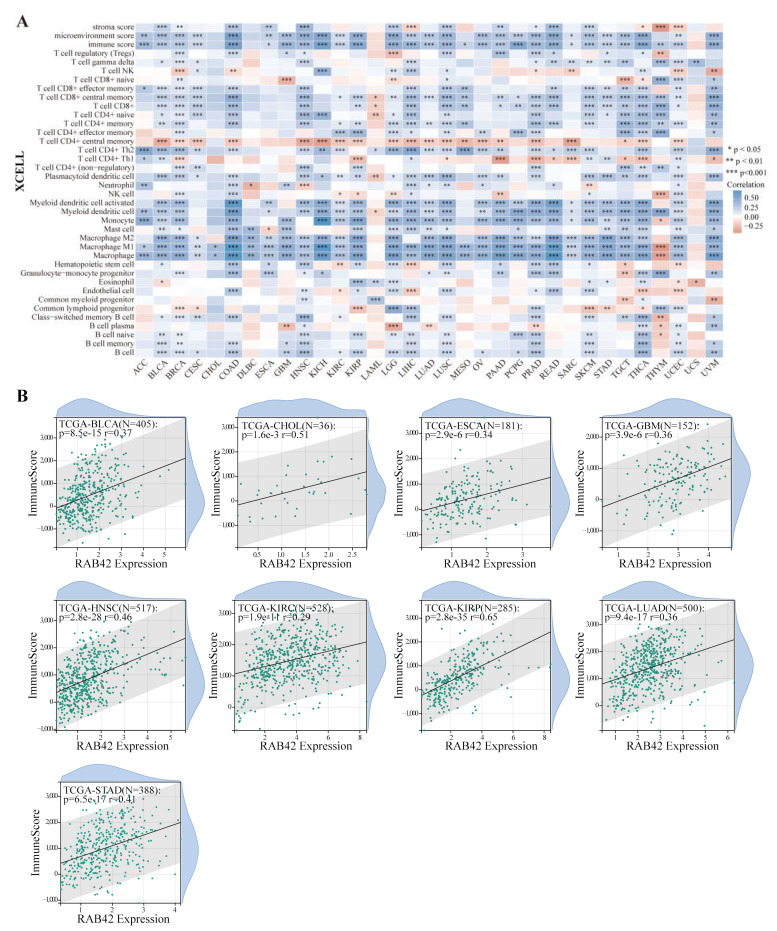
Correlation analysis between RAB42 expression and immune cell infiltration. (**A**) Spearman’s correlation between RAB42 expression and immune cell infiltration in pan-cancer. (XCELL algorithm); (**B**) Spearman’s correlation between RAB42 expression and immune score. * *p* < 0.05, ** *p* < 0.01, and *** *p* < 0.001.

**Figure 7 biomolecules-13-00043-f007:**
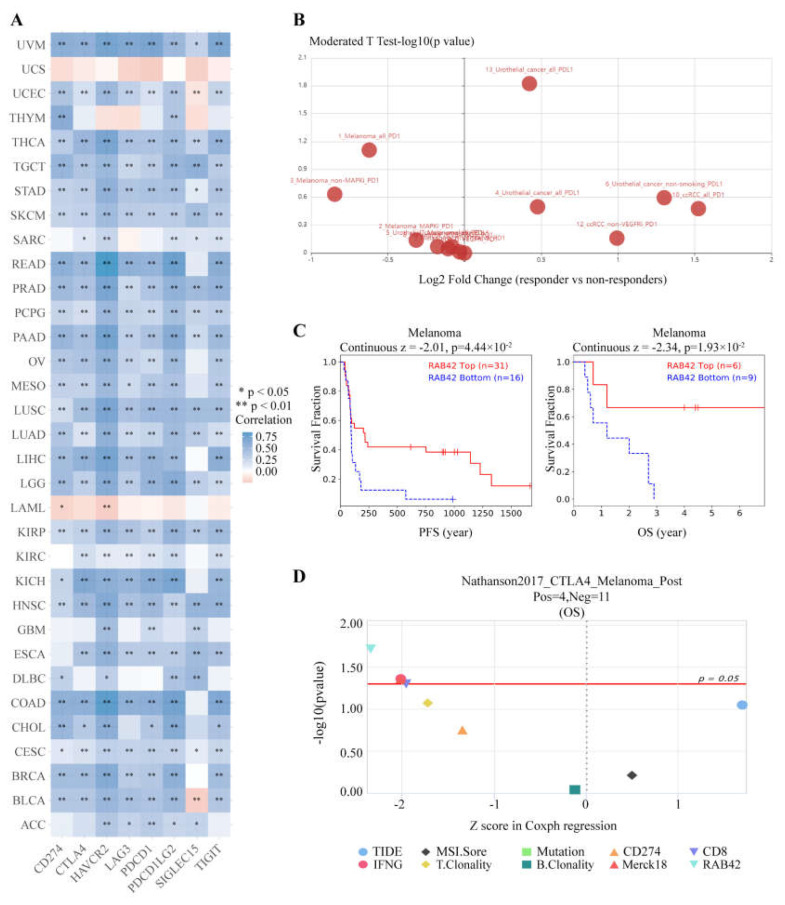
RAB42 was a predictor for immunotherapy. (**A**) Spearman’s correlation between RAB42 and immune checkpoint genes expression in the ACLBI database. (**B**) Differences in RAB42 expression among responders and non-responders treated with PD1/PD-L1 inhibitors using the TISIDB database. (**C**) The effect of RAB42 expression on the prognosis of melanoma patients treated with PD1 or CTLA4 inhibitors using the TIDE database. (**D**) Predictive value of RAB42 for immunotherapy using the TIDE database. * *p* < 0.05; ** *p* < 0.01.

**Figure 8 biomolecules-13-00043-f008:**
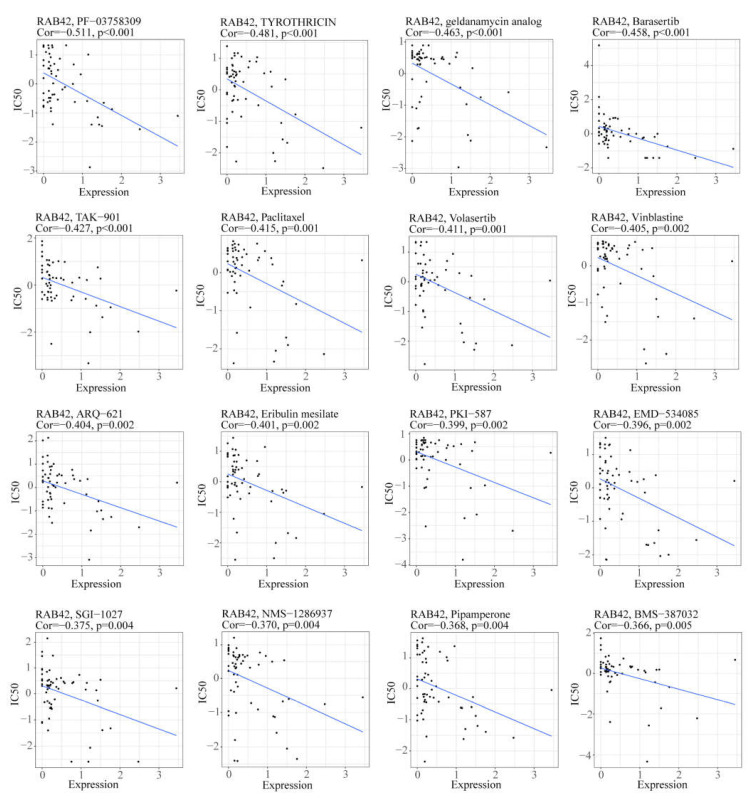
The scatter plots illustrated the correlation between RAB42 expression and IC50 of drugs.

**Figure 9 biomolecules-13-00043-f009:**
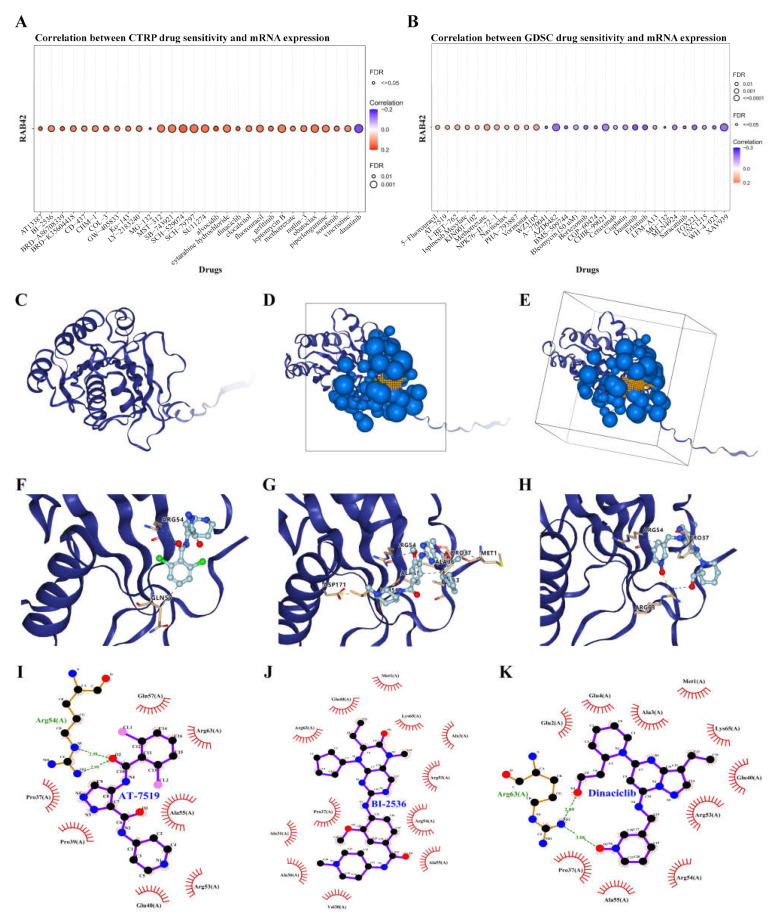
Screening susceptible drugs for the RAB42 protein. (**A**,**B**) Predicted drugs sensitive for RAB42 by the GDSC and CTRP datasets. (**C**) The 3D structure of the RAB42 protein. (**D**,**E**) The binding site and box of the RAB42 protein. Small orange spheres represent binding site, black solid lines represent binding box. (**F**–**H**) The 3D docking conformations of AT-7519, BI-2536 and Dinaciclib with the RAB42 protein. Labels represent amino acid residues; blue dashed line represents hydrogen bond; yellow dashed line represents salt bridge (ionic bond); gray dashed represents hydrophobic interaction. (**I**–**K**) The interaction forces between amino acid residues of the RAB42 protein and AT-7519, BI-2536 or Dinaciclib. Green dashed line represents hydrogen bonds.

**Figure 10 biomolecules-13-00043-f010:**
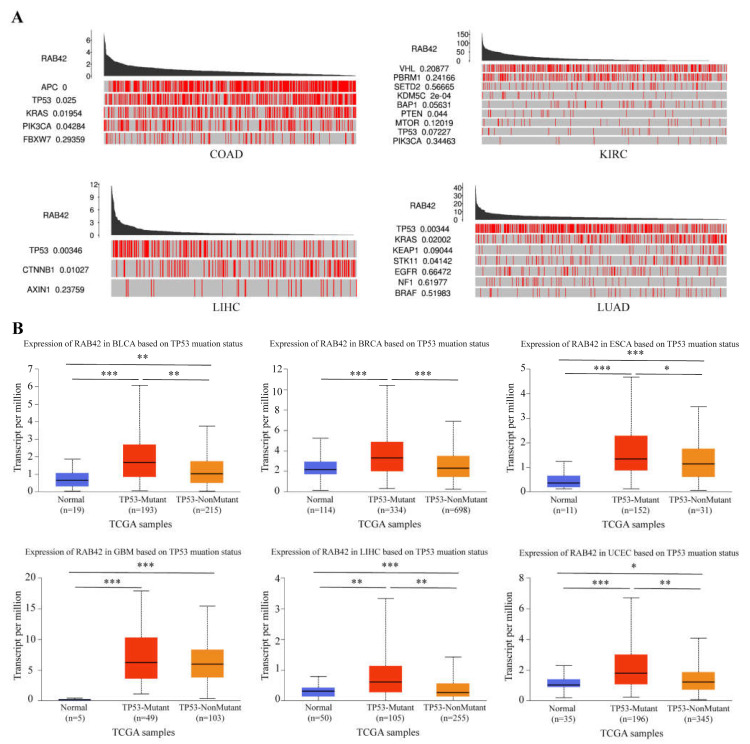
The effect of driver genes on RBA42 expression. (**A**) The correlation between driver genes and RAB42 expression using the TCGA Portal database. (**B**) The effect of TP53 mutation status on RAB42 expression using the UALCAN database. * *p* < 0.05, ** *p* < 0.01, and *** *p* < 0.001.

**Figure 11 biomolecules-13-00043-f011:**
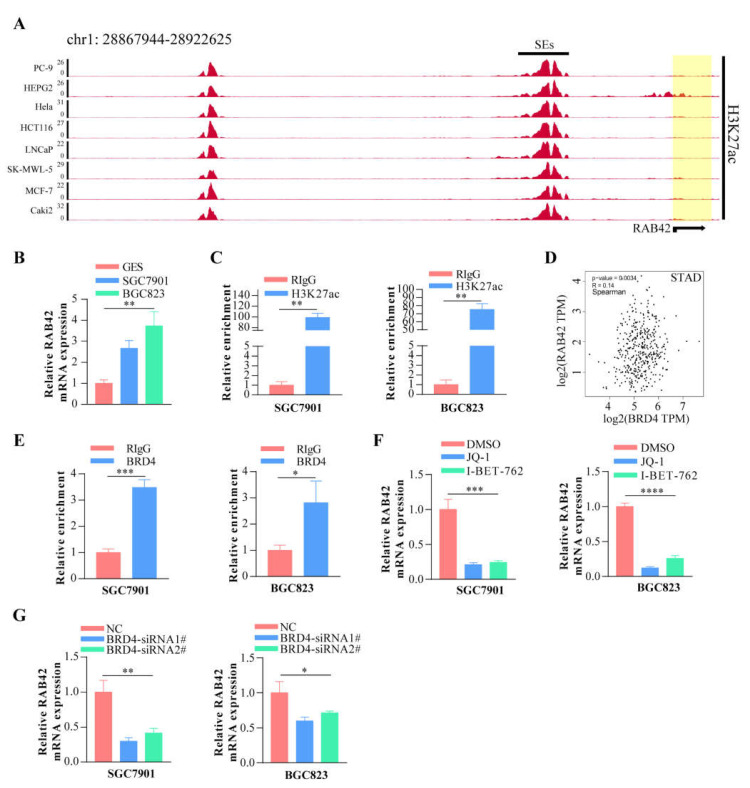
The effect of super enhancer on RAB42 expression. (**A**) ChIP-seq profiles of H3K27ac signal in RAB42 gene loci in different tumor cells. (**B**) The relative RAB42 expression was deter-mined in gastric cancer cells by qPCR. (**C**) The relative enrichment levels of H3K27ac to RAB42 gene loci by ChIP-qPCR. (**D**) Correlation between BRD4 and RAB42 expression in STAD using the GEPIA database. (**E**) The relative enrichment levels of BRD4 to RAB42 gene loci by ChIP-qPCR. (**F**) The relative RAB42 expression after the treatment with JQ1 (1uM) and I-BET-762 (2uM) by qPCR. (**G**) The relative RAB42 expression after BRD4 knockdown by qPCR. **p* < 0.05, ***p* < 0.01, ****p* < 0.001, and **** *p* < 0.0001.

**Figure 12 biomolecules-13-00043-f012:**
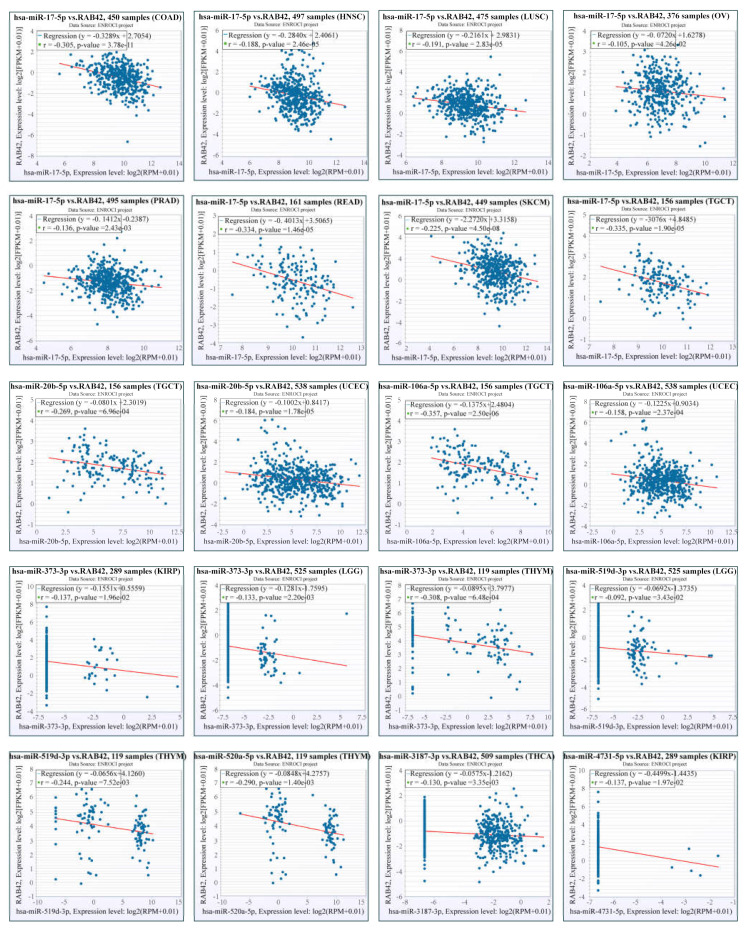
Co-expression analysis for the miRNA-RAB42 interactions.

## Data Availability

Publicly available datasets were analyzed in this study. These data can be found in online databases. Please refer to the method section for how to use these databases.

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
