# Peer review of "Pan-Cancer Analysis of the Roles and Driving Forces of RAB42"

_biomolecules, 2022, doi:10.3390/biom13010043_

Round 1

Reviewer 1 Report

This manuscript analyze the role of Rab42 in cancer and I find the information very interesting and relevant for future strategies on cancer therapeutics. There are, however, several aspects in which the manuscript can be improved. The text must follow an extensive revision of the English language, some of the suggestions for improvement are marked in the accompanying file, but someone more proficient in English should check the manuscript before publication. 

A major improvement of this work would be an analysis of the relationship between the different microRNA (lncRNA) presented in the last figure, their alteration in cancer and the levels of Rab42 expression. While interesting, this list of microRNA (lncRNA) do not add information to the data associating Rab42 and cancer and that association should be checked by the authors.

Another point of improvement would be a list of abbreviations. There are too many and they are not explained anywhere, so either the reader is an expert oncologist or many of the abbreviations for the different cancer types have to be searched elsewhere.

Some figures have too much information and loose quality. A lower number of panels with the most representative alterations would help to present the data, meanwhile, the rest of the information can be offered as supplementary material.

Other improvements to this manuscript can be found as annotations in the attached file.

Author Response

  1. The text must follow an extensive revision of the English language, some of the suggestions for improvement are marked in the accompanying file, but someone more proficient in English should check the manuscript before publication.

Reply: We revised our manuscript according to the reviewer’ suggestions and improved the quality of the writing with the help of a foreign friend.

  1. A major improvement of this work would be an analysis of the relationship between the different microRNA (lncRNA) presented in the last figure, their alteration in cancer and the levels of Rab42 expression. While interesting, this list of microRNA (lncRNA) do not add information to the data associating Rab42 and cancer and that association should be checked by the authors.

Reply: We added the relevant analysis according to the reviewer’ suggestions (Figure 12, Figure S9, Figure S10 and Table S4).

  1. Another point of improvement would be a list of abbreviations. There are too many and they are not explained anywhere, so either the reader is an expert oncologist or many of the abbreviations for the different cancer types have to be searched elsewhere.

Reply: All the abbreviations have been indicated in full within the text, where they are first mentioned. And we added a list of abbreviations in Table S1.

  1. Some figures have too much information and loose quality. A lower number of panels with the most representative alterations would help to present the data, meanwhile, the rest of the information can be offered as supplementary material.

Reply: We revised our figures according to the reviewer’ suggestions. And we provided new figures with good quality.

  1. Other improvements to this manuscript can be found as annotations in the attached file.

Reply: We revised our manuscript according to the annotations in the attached file.

Reviewer 2 Report

Zheng et al. used online databases to explore statistical correlations between RAB42 and various cancer types, evaluated potential drug targets for RAB42, as well as potential genetic and epigenetic factors regulating RAB42 expression. This is an interesting study categorizing effects of RAB42 in cancer.

Major issues:

The authors screened potential drug targets for RAB42 (Pages 11 and 12). It is notorious that small GTPases are very difficult to inhibit. As such, the likelihood that the molecules identified in the current study can actually inhibit activities of RAB42 is very slim to none.

In the next section (Pages 13-16), the authors explored genetic and epigenetic factors regulating RAB42 expression. Unlike many receptor tyrosine kinases, small GTPases are not over-expressed in diseases. On the other hand, many small GTPases undergo oncogenic mutations. Have the authors explored RAB42 mutations commonly associated with various cancer types?  

Author Response

1.The authors screened potential drug targets for RAB42 (Pages 11 and 12). It is notorious that small GTPases are very difficult to inhibit. As such, the likelihood that the molecules identified in the current study can actually inhibit activities of RAB42 is very slim to none.

Reply: The reviewer is very professional. Small GTPases are indeed very difficult to inhibit. This study aimed to explore the characteristics of RAB42 from a pan-cancer perspective. Our study revealed the critical oncogenic roles of RAB42 in pan-cancer. Therefore, we screened out susceptible drugs for RAB42 protein by bioinformatic analysis and virtual screening. Our study aimed to provide theoretically possible drugs. Of course, the effects of these drugs need to be verified by specific in vitro and in vivo experiments.

2.In the next section (Pages 13-16), the authors explored genetic and epigenetic factors regulating RAB42 expression. Unlike many receptor tyrosine kinases, small GTPases are not over-expressed in diseases. On the other hand, many small GTPases undergo oncogenic mutations. Have the authors explored RAB42 mutations commonly associated with various cancer types? 

Reply: We have explored RAB42 mutations in Figure S7A. The results indicated that RAB42 generally showed low mutational frequencies in pan-cancer. Mutations may be not the primary cause for RAB42 oncogenic roles.